# Molecular Investigations of *Babesia caballi* from Clinically Healthy Horses in Southwestern Romania

**DOI:** 10.3390/vetsci11120600

**Published:** 2024-11-27

**Authors:** Simona Giubega, Marius Stelian Ilie, Sorin Morariu, Mirela Imre, Cristian Dreghiciu, Tatiana Rugea, Simina Ivascu, Gheorghița Simion, Gheorghe Dărăbuș

**Affiliations:** 1Department of Parasitology and Parasitic Disease, Faculty of Veterinary Medicine, University of Life Sciences “King Mihai I” of Timisoara, 119, Calea Aradului, 300645 Timisoara, Romania; simonagiubega@gmail.com (S.G.); sorin.morariu@fmvt.ro (S.M.); mirela.imre@usvt.ro (M.I.); cristian.dreghiciu@usvt.ro (C.D.); gheorghe.darabus@fmvt.ro (G.D.); 2Veterinary and Food Safety Department 4, Surorile Martir Caceu, 300585 Timisoara, Romania; rugea.tatiana-tm@ansvsa.ro (T.R.); ivascu.elena-tm@ansvsa.ro (S.I.); simion.gheorghita-tm@ansvsa.ro (G.S.)

**Keywords:** *Babesia caballi*, equine piroplasmosis, real-time PCR

## Abstract

*Babesia caballi* is a tick-borne parasite that affects horses. It can cause illness and economic problems. This study aimed to find *B. caballi* DNA in the blood of 310 horses in southwestern and western Romania. Blood samples were tested using two different methods. The results showed that this parasite is common in healthy horses in the area, even when they do not show signs of babesiosis. This study shows that we do not have enough information about equine babesiosis in Romania. We must conduct more research to understand how it spreads and find ways to stop it.

## 1. Introduction

Equine piroplasmosis (EP) is caused by *Babesia caballi* and *Theileria equi,* two intraerythrocytic apicomplexan hemoprotozoa. It is a tick-borne disease transmitted by ticks of the genera *Dermacentor*, *Hyalomma*, and *Rhipicephalus* [1,2,3]. A third species responsible for EP, *Theileria haneyi*, has recently been discovered in North America but has not been described in Europe so far [4]. Babesiosis is endemic in many parts of the world, in countries in Asia, Africa, the Americas, and especially Europe [1,2,3]. *Babesia caballi* is usually not prevalent in most areas, with generally lower parasitemia and a milder clinical course than *T. equi* [1]. The clinical signs of EP include fever, edema, anemia, jaundice, hemoglobinuria, and low performance. Superacute or acute forms of infection are associated with high mortality, mainly in non-endemic areas, and this can reach up to 10% of infected horses [5].

As the geographical range of many tick-borne diseases is expanding because of climate change, it is essential to identify the diversity of tick species, especially in correlation with their host associations. In Europe, several species of ixodid ticks belonging to the *Hyalomma*, *Dermacentor*, and *Rhipicephalus* genera have been reported, these being the most well-known and studied vectors involved in transmitting piroplasmas [6]. Other tick genera from different parts of the world, such as *Amblyomma*, *Haemaphysalis*, and *Ixodes*, are suspected to be vectors, but studies on their vectorial capacity are insufficient [6,7]. In Europe, the most prevalent tick species feeding on equids are *Hyalomma marginatum*, *Ixodes ricinus*, *Rhipicephalus bursa*, and *Dermacentor reticulatus* [4]. Ticks are a reservoir for many pathogens in general and *Babesia* spp. in particular, including *B. caballi*. An important element in the persistence of infection in the tick population is the trans-stadial and transovarial transmission of the protozoan. [7]. Most tick species are widespread throughout Romania, and only a few are found in specific areas. *Dermacentor marginatus* is widespread throughout the country, with a higher prevalence in the west and southeast; *D. reticulatus* is more common in the northeastern part of the country and *H. punctata* in the west, while *H. marginatum* is more common in the southeast, south, and west [8].

*Babesia caballi* is reported to be endemic in most parts of the world, including Europe, but in Romania, the data are lacking. So far, 25 species of ticks belonging to the *Ixodidae* family have been identified in our country, which has identified the species involved in transmitting piroplasma, as well as the habitats that are suitable for the development of these vectors, which represent approximately 50% of the country’s surface [8,9].

The main diagnostic method for identifying this species is blood smear microscopy, which is useful for detecting acute infections but not for detecting carriers with very low parasitemia or animals that have already experienced the disease [10]. Because of the difficulties of microscopically identifying parasites in carrier or low-parasitemia animals, serologic methods such as the complement fixation test (CFT), the indirect fluorescent antibody assay (IFAT), and the enzyme-linked immunosorbent assay (ELISA) have been developed to aid the diagnostic process [11]. While ELISA is used to detect antibodies, the real-time PCR method is useful for detecting parasites in the vertebrate host. Horses infected with *B. caballi* have detectable levels of antibodies 3–21 days postinfection [12], but nothing is known about titer persistence [13]. According to the WOAH Terrestrial Manual 2021, both cELISA and PCR are recommended for testing the animal population that is free of infection in order to contribute to eradication policies, as well as infection (prevalence) surveillance; however, only PCR is recommended for confirmation of clinical cases, and cELISA is not suitable for this purpose [14]. Molecular methods, such as PCR, are much more sensitive than light microscopy and can be useful for detecting *B. caballi* in the carrier state or during chronic infection when parasitemia is very low [15]. Because seropositive animals in an asymptomatic population are not indicators of recent or active infection, testing via molecular methods is the only choice for confirming or disproving the presence of the parasite’s genomes [16]. To date, numerous diagnostic protocols have been studied and are available, using molecular biology methods to target genes such as 18S rRNA [17], the BC48 (Merozoite Rhoptry Protein) gene [18], RAP-1 (rhoptry-associated protein) [17], and β-tubulin [19].

The therapeutic approach varies according to the epidemiological status. In endemic regions, treatment goals include improving clinical signs and preventing death, whereas, in non-endemic regions, the treatment goal is to completely eradicate the infection to prevent the spread of parasites to other horses. Imidocarb dipropionate is the drug of choice for treating equine piroplasmosis and is considered the safest of all available drugs, with high efficacy in treating clinical cases [20,21]. Treatment in the acute phase is largely supportive, and it includes imidocarb dipropionate, the treatment of choice for parasite clearance. The premunition phenomenon (the stage in a disease where an existing infection protects the host from reinfection with the same pathogen) protects animals in endemic areas, and in non-endemic areas, it is very important to eliminate any risk of transmission to other animals [1]. *Babesia caballi* represents a major problem in horse health and has a significant economic impact, especially in terms of animal movement and international trade [22]. This pathology frequently occurs in rural areas of developing countries, including Romania, where horses are still used for work, e.g., agriculture [23].

In Europe, numerous studies have been carried out with positive PCR results for *B. caballi* in countries such as The Netherlands, Poland, Hungary, Romania, Croatia, Italy, and the central Balkans [24]. In Romania, epidemiologic data on its prevalence are lacking or insufficient, and this study aimed to identify the presence of this parasite’s genome in horses in households in the southwestern and western parts of Romania, areas where horses are mainly used for agriculture and kept free-range on pastures.

## 2. Materials and Methods

This study included 310 animals (n = 172 females and n = 138 males) from households. According to the National Institute of Statistics, the total number of horses in Romania in 2023 was 290,246, with 44,054 in the southwestern region and 47,164 in the southwestern and western regions [25]. The total number of horses that were sampled was determined using the epiR function of the R package [26], with a 5% expected prevalence, a 95% confidence interval (CI), and 99% diagnostic sensitivity. Thus, the required number of horses was 61. In this study, the number of horses was increased to improve the precision of the estimated prevalence. The horses included in the study were selected randomly. Blood samples from equine animals were collected in sterile EDTA tubes between August 2022 and April 2024 from the southwestern, western, northwestern, and northern areas of Romania from the counties of Gorj (n = 91), Mehedinți (n = 15), Caras-Severin (n = 36), Timiș (n = 162), Satu Mare (n = 4), and Bistrița Năsăud (n = 2) (Figure 1). If it was not possible to process them in a short time, they were stored at −80 °C until examination.

### 2.1. DNA Extraction

Genomic DNA was extracted from 200 μL of blood with EDTA using the MagMAX™ CORE Nucleic Acid Purification Kit (Applied Biosystems™, Waltham, MA, USA), according to the manufacturer’s recommendations, and the KingFisher™ Flex Purification System (Thermo Scientific™, Waltham, MA, USA).

### 2.2. DNA Quantification

DNA levels vary depending on the quality and number of cells in the sample, so for whole blood, optimal levels are between 8 and 50 ng/µL of blood [16]. The amount of nucleic acid extracted was quantified using a Qubit^®^ 4.0 fluorometer and the Qubit^®^ dsDNA HS Assay Kit. The Qubit^®^ dsDNA HS Assay used 10 µL of the DNA sample + 190 µL of working solution. The same dilution was used for the two standards. Quantification was carried out to check the extraction process and to verify that there was not too much DNA in the sample, which could inhibit the real-time PCR reaction, i.e., increasing the possibility of obtaining a false negative result. The amount of DNA per sample was 8.5 µL, together with 16.5 µL of a master mix, providing a final reaction volume of 25 µL. With this volume, we approached the optimal values, the main goal being to avoid having too much DNA.

### 2.3. Real-Time PCR

The samples were subjected to the protocol described by Bhoora (2010) [17] with a TaqMan^®^ MGB™ (minor groove binder) probe, a Bc_18SP (5′-6-FAM-CCT CGC CAG AGT AA-MGB-3′) probe, and a pair of primers (Bc_18SF402: 5′-GTA ATT GGA ATG ATG GCG ACT TAA-3′; Bc_18SR496: 5′-CGC TAT TGG TGG AGC TGG AAT TAC C-3′). The probe and primers were designed to detect a 95 bp fragment in the V4 hypervariable region of the 18S rRNA gene. The forward primer and probe were specific to *B. caballi*, but the reverse primer was not [17].

The certificate of analysis accompanying the primers and probe contained details on the amount and concentration; the storage and aliquoting concentration was 100 μM, and the working concentration was 10 μM for the primers and 3 μM for the probe.

The amplification was carried out using a QuantStudio™ 7 Flex Real-Time PCR System (Applied Biosystems™) under the following thermal cycling conditions: initial denaturation at 95 °C for 10 min and 40 cycles of thermocycling, including denaturation at 95 °C for 15 s and annealing and extension at 60 °C for 45 s each. After each sample was tested, the Ct value was determined via the log-linear phase of each reaction. The results were interpreted on the basis of a standard Ct relative to the number of amplification cycles, and the threshold was automatically selected by the PCR platform so as not to influence the result.

The following additional controls were added to each reading cycle for plate validation: ME (negative extraction control—water or a known safe negative sample) and NTC (no template control—amplification mix + ultrapure water). Figure 2 shows the Ct values of the positive samples plotted against their mean value of 29.74083. In fact, the results were expressed on the basis of an analysis of the amplification curves and the Ct value of amplification; thus, samples with Ct ≤ 40 were considered to be positive and samples with Ct > 40 were considered to be negative.

### 2.4. Conventional PCR

For sequencing, five randomly chosen positive samples were tested via conventional PCR, amplifying a 95 bp segment and using a total reaction volume of 25 µL. The protocol comprised 5 μL of DNA, 12.5 μL of MyTaqTM Red Mix (BIOLINE^®^, Memphis, TN, USA), 1 μL of the primer Bc_18SF402, 1 μL of the primer Bc_18SR496, and 5.5 μL of ultrapure water. Amplification was carried out with a My Cycler thermocycler (Bio-Rad^®^, Hercules, CA, USA) via a 32-cycle amplification program, i.e., initial DNA denaturation at 95 °C for 1 min, followed by denaturation at 95 °C for 30 s, alignment at 58 °C for 30 s, and extension at 72 °C for 30 s, then followed by incubation at 4 °C to allow the PCR products to fully extend. In the PCR reaction, the positive control and a negative control consisting of ultrapure water were included. The positive control was one of the first samples identified by real-time PCR. Amplicon analysis was performed via horizontal electrophoresis in a 1.5% agarose gel submersion system with the addition of MidoriGreen fluorescent dye (Nippon Genetics^®^ Europe, Mariaweilerstraße, Duren, Germany).

### 2.5. Sequence Analysis

The PCR products for 5 samples were purified using Alphagen’s α+ Solution™ GEL/PCR Purification Kit, according to the manufacturer’s recommendations, and sent for sequencing to Macrogen, Amsterdam, The Netherlands. The obtained sequences were edited and compared with sequences available in GenBank using NCBI BLAST [27].

### 2.6. Statistical Analyses

Statistical analysis was performed based on categories according to sex (2 groups: males and females) and age (3 categories: young animals, 0–60 months; adults, >60–180 months; old animals, >180 months). The positivity in the real-time PCR assay was compared among the different age and sex groups by applying Fisher’s test and the Chi-square test in GraphPad Prism 9.2.0. A *p*-value of <0.05 was considered significant.

## 3. Results

### 3.1. DNA Extraction and Quantification

After DNA extraction from 310 blood samples taken from clinically healthy horses, DNA quantification was performed on 50 randomly selected samples. The quantified values ranged from 0.68 to 9.82 ng/µL. DNA levels vary depending on the quality and number of cells in the sample, so for whole blood, optimal levels are between 8 and 50 ng/µL of blood [28].

### 3.2. Real-Time PCR

Following the partial amplification of the 18S rRNA gene, *B. caballi* DNA was detected in 18 samples (5.81%) (Table 1). The Ct values of the positive samples are presented in Figure 2 and ranged from 24,582 to 38,415, but the Ct value indicates the current state at the time of testing and should be taken in the context of the symptoms and the clinical course. In particular, a Ct value of 24,582 does not necessarily indicate a clinical expression of the disease, although parasitemia is higher than that in a positive animal with a Ct of 38,415. The lower the Ct value, the more DNA is in the initial sample. On the other hand, high Ct values, closer to the detection limit of the assay (often ~40 cycles), indicate small amounts of DNA in the test sample. A unit change in the Ct value represents an exponential increase in (doubling of) the target.

Among the total positive samples, 12 were from females (6.97%) and 6 were from males (4.34%). In terms of age category, 1 belonged to the 0–60-month age category (1.26%, 1/79), 11 to the >60–180-month age category (7.58%, 11/145), and 6 to the >180-month age category (6.97%, 6/86). The youngest positive horse was 20 months old, and the oldest was 277 months old.

The positive results came from two of the six counties surveyed, namely Timis (4/162) and Gorj (14/91). No animals were positive in the other regions. A data analysis revealed that the horses from Gorj County had the highest infection rate (15.38%).

The results of the statistical analysis revealed no significant differences, with a *p*-value of 0.4657 for the two sex categories; for the categories resulting from the age division, the *p*-value was 0.0639 for the 0–60-month/>60–180-month categories, with *p =* 0.7909 for the >60–180-month/>180-month categories, and *p* = 0.2163 for the 0–60-month/>180-month categories.

### 3.3. Conventional PCR

The five randomly selected positive samples tested using conventional PCR amplified a 95 bp segment. Amplicon analysis was performed via horizontal electrophoresis in a 1.5% agarose gel immersion system with the fluorescent dye MidoriGreen, and all samples tested were positive (Figure 3).

### 3.4. Sequence Analysis

The sequences were deposited in GenBank with the accession numbers SUB14469064 Seq1 PQ317200, SUB14469064 Seq2 PQ317201, SUB14469064 Seq3 PQ317202, and SUB14469064 Seq4 PQ317203. The sequencing results revealed *Babesia caballi* with 98% to 100% similarity to several GenBank isolates, such as KF055854.1 (China), MT355491.1 (China), MN907451.1 (China), MN907450.1 (China), and MN629354.1 (Israel).

## 4. Discussion

Romania is one of the few countries where there is insufficient or no information on the epidemiology of EP, denoting a lack of knowledge on the circulation of piroplasma species and their vectors. This represents a risk for susceptible animal populations in the country and the rest of Europe. An epidemiological study was necessary to highlight the prevalence and circulation of this parasite in Romania. In the study areas, competent vectors for *B. caballi* infection have been previously identified, namely ticks of the genera *Dermacentor*, *Hyalomma*, and *Rhipicephalus* [8,9]. The potential vectors of EP were identified following a study by Coipan et al., which showed that *Hyalomma* spp. are most common in the southeastern area, while *Rhipicephalus* is most common in the south and *Dermacentor* in the west [8]. Factors such as global warming, animal movement, the intensive use of rural areas, and the ways animals are used have contributed to and increased the spread of several tick-borne pathogens. In the European Union (EU), after the Second World War, the motorization of transport led to a drastic decline in the number of Equidae by around 90% in the 1950s [29]. Romania was no exception, with the number of Equidae falling from 670,000 in 1990 to 290,246 in 2023 [25].

In Romania, the first study on the prevalence of EP was conducted by Gallusová et al. from 2010 to 2012, showing a prevalence of 25.4% for both piroplasma species, whereas the prevalence for *B. caballi* was 4.5% (8/193) [30]. Studies have not been conducted in these areas; thus, the epidemiologic situation is currently unknown. Our research focused on animals that are currently used in agriculture in rural households. The real-time PCR prevalence of *B. caballi* was 5.81% (18/310), which is close to that reported by Gallusova et al. but higher than the estimated overall prevalence in Europe [4]. A prevalence of 2.4% and a seroprevalence of 7.8% have been reported for *B. caballi* in Europe. The different results between the two studies carried out in Romania may be due to differences in the study design (i.e., the methods used, real-time PCR, and conventional PCR) but also to differences in the type of climate, environment, vectors, and number of animals in the two areas.

Mixed grazing, a common practice in Romania, is considered a risk factor for *B. caballi* infection, accounting for the treatment of different species. An important factor in the differences in prevalence is the way horses are used and, more specifically, the diversity of their use. This has not only led to debates about the status of Equidae (as livestock or pets) between European countries but also within each country. For example, in the United Kingdom, horses are considered to be pets, whereas in other European countries, such as Romania, France, Germany, and Sweden, they are considered to be farm animals [29].

Although most of the animals were reported to be in good health and without symptoms similar to those of *B. caballi* infection, this parasite was detected in their blood samples; the results differed from those of a study published by Ioniță et al., who reported the first molecular confirmation of this parasite, in which *B. caballi* DNA was sequenced from four clinically affected horses [31]. This observation demonstrates the potential risk of infection. This parasite’s dynamics of transmission in horse populations should be further investigated, and potential prevention and control strategies should be identified. The age of the horses can be a risk factor due to long exposure to the tick vectors; although infection was higher in the >60–180-month age group, no statistical differences between positivity and the age of the animals were found. These differences may be due to several factors related to the long-term circulation of the parasite’s host, as well as to host–parasite interactions. While studies in Europe have identified sex as a risk factor for *B. caballi* infection, in the present study, the sex-related prevalence was found to be statistically insignificant (*p* = 0.4657).

In Europe, positive results for *B. caballi* in PCR tests have been reported in Italy (10.3%) [16], Croatia (3.6%) [32], the central Balkans (2.1%) [33], and France, with the authors detecting a prevalence of 6.3% using real-time PCR in horse blood [34]. Numerous studies have reported results contrary to those of the present research, i.e., a prevalence of 0% for *B. caballi* in countries such as The Netherlands [35], Hungary [36], and Poland [37], in which *B. caballi* DNA was not detected via PCR or dual infection. However, Butler et al. found that one horse was positive for the *Babesia* catch-all genus [35]. Different results from those of the present research have also been reported in Serbia, where a study on 70 apparently healthy donkeys showed that no animal was positive for *B. caballi* [38]. In 2021, Dirks et al. published a study revealing that in Austria, no autochthonous case of *B. caballi* infection has been recorded [13]. At the European level, Spain, Italy, and France are the most affected by PD, with a prevalence between 1 and 5%, possibly because of the number of studies carried out in these countries: 9 in Spain and 10 in Italy [4].

Although our results agree with many studies published to date, some studies have reported a significantly higher prevalence, such as in Brazil, where the results of a study on 516 animals revealed a 17.2% prevalence [39]. Considerable differences have been reported between serological and molecular methods. Bělková et al. published a study carried out in the Czech Republic, reporting a 0.4% seroprevalence for *B. caballi* but no PCR-positive results for serologically positive horses [40]. Tirosh-Levy et al. reported a seroprevalence of 69.6% and a molecular prevalence of 9.7% [41]. There are countries where equine piroplasmosis is believed to be underdiagnosed, such as Switzerland, where it is considered a sporadic disease [42], although in a study of 689 horses, a prevalence of 2.9% was reported for *B. caballi* [43]. The resulting prevalence in our study is similar to that reported by Rosales et al. (4.4%) and Salinas-Estrella (7.8%) [44,45]; moreover, a study in which animals were tested via real-time PCR reported a prevalence of 6.3% [34]. In most articles published in the literature, the prevalence of *B. caballi* has been reported to be much lower than that of *T. equi* [34,46,47,48,49,50], which may be due to the nature of *T. equi* infections. *B. caballi* is usually self-limiting and can persist for up to four years without clinical signs, after which it is naturally cleared [20,51,52,53,54]. In addition, the parasites’ sequestration in bone marrow results in parasitemia that may not exceed 1% in the blood of naturally infected horses [20,54].

The lack of clinical signs is usually correlated with equine populations in endemic areas, as early exposure is likely to induce protection [15]. However, clinical cases have been reported in Romania [31], and this may also be due to differences in the vector exposure of different equine subpopulations. The fact that *B. caballi* infection responds to imidocarb treatment, together with its ability to self-eliminate from the body [11], may contribute to limiting its spread in most areas, with lower parasitemia and milder clinical expression than those of *T. equi* [41]. Owing to differences in the sensitivity of diagnostic tests used in epidemiological studies, differences in prevalence have been reported from different areas of the world. These differences in prevalence may be due to risk factors that favor infection, such as the presence and abundance of competent vectors, host activity, management practices, and the effectiveness of vector control programs, if any. These results have opened new horizons of research. They were generated according to the need to further explore this pathogen’s transmission mechanisms, the vectors involved in its carriage, and its natural reservoirs, as it may go unnoticed once animals pass the acute stage.

## 5. Conclusions

This present study contributes to understanding the spread of *B. caballi* in the southwestern and western regions of Romania, as well as the fact that it is endemic in the studied regions. However, veterinary practitioners have not reported a similar piroplasmosis-like symptomatology. This parasite represents a significant threat to horse health, with chronically infected animals representing a reservoir for the spread of the disease. A prevalence of 5.81% identified in apparently healthy horses suggests the susceptibility of animals to this parasite, as well as the need for further investigations on the dynamics of transmission and to identify prevention and control strategies for Romania and Europe.

## Figures and Tables

**Figure 1 vetsci-11-00600-f001:**
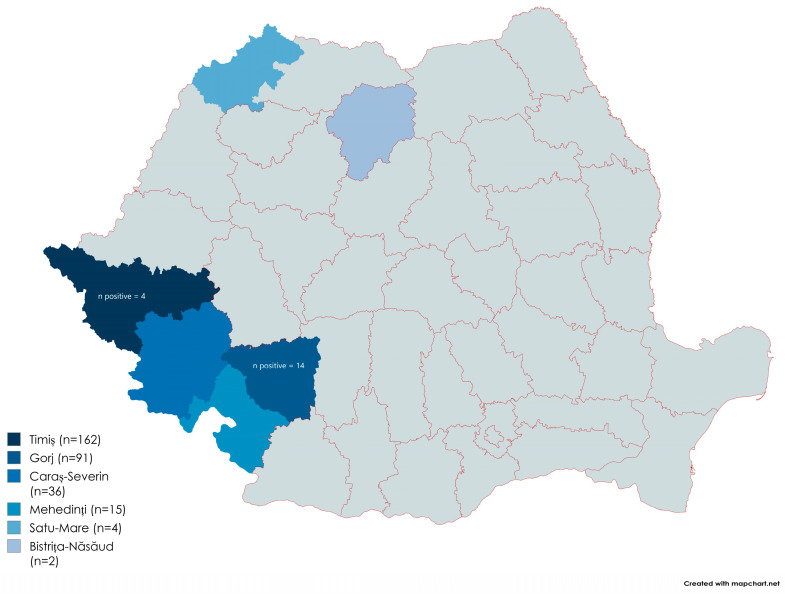
Origin of the samples by county.

**Figure 2 vetsci-11-00600-f002:**
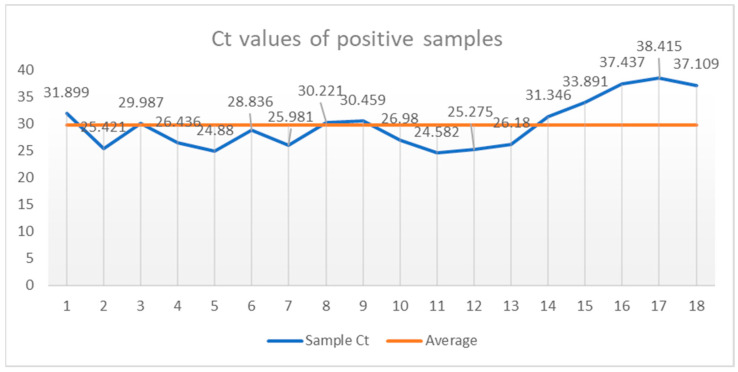
Cycle threshold values in positive samples (blue) compared with their mean (orange) (29.74083).

**Figure 3 vetsci-11-00600-f003:**
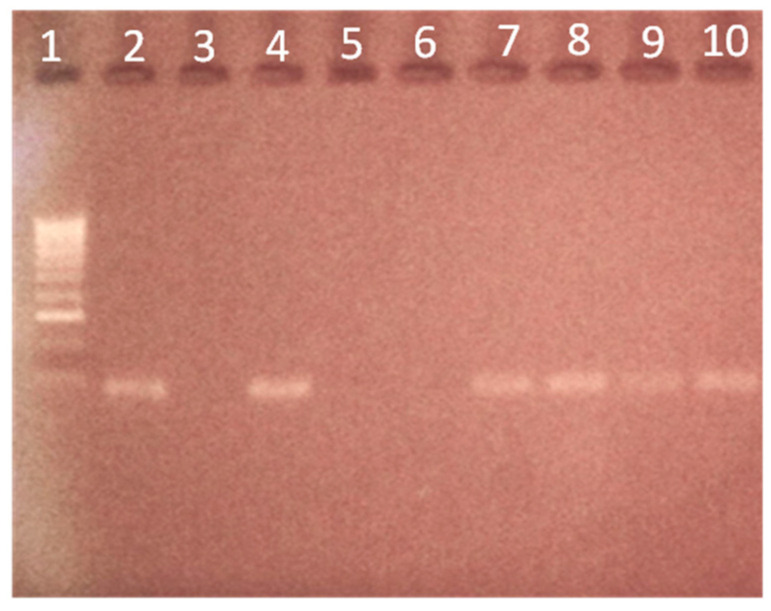
PCR gel electrophoresis of the 18S rRNA gene of *B. caballi* isolates. Line 1, 100 bp DNA ladder (BIOLINE^®^ UK Ltd., London, UK); Line 2, positive control (*B. caballi*); Line 3, negative control; Lane 4, positive sample; Lines 5–6, negative samples; Lines 7–10, positive samples.

**Table 1 vetsci-11-00600-t001:** Distribution of positive horses according to the studied risk factors.

Factor	No. of Equines Tested	N* (%) *	*B. caballi* OR * (95% CI *)
	310	18 (5.81%)	0.0581 (0.0370–0.0899)
Sex			
Female	172	12 (6.97%)	0.0698 (0.0404–0.1180)
Male	138	6 (4.34%)	0.0455 (0.0205–0.1008)
Age group			
0–60 months	79	1 (1.26%)	0.0127 (0.0022–0.0683)
60–180 months	145	11 (7.58%)	0.0759 (0.0429–0.1307)
>180 months	86	6 (6.97%)	0.0698 (0.0324–0.1440)

* N, number of positive samples; %, prevalence; OR, odds ratio; CI, confidence interval; *p*, *p*-value.

## Data Availability

The original contributions presented in the study are included in the article, further inquiries can be directed to the corresponding author.

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
