# Peer review of "Molecular Investigations of *Babesia caballi* from Clinically Healthy Horses in Southwestern Romania"

_vetsci, 2024, doi:10.3390/vetsci11120600_

Round 1
Reviewer 1 Report
Comments and Suggestions for Authors
I have following suggestions.
Introduction
Many short sentences were listed mechanically. I suggest to make two or three paragraphs such as etiology of EP including vectors, diagnostic methods, epidemiology of world-wide and Romania, and aim of present study.
Line 24: Suggesting to change “Equine piroplasmosis (EP).
Line 27: Suggesting to change “Equine piroplasmosis” from “Babesiosis”.
Results
Suggesting to have subtitles of results.
Suggesting to add a phylogenetic tree according by the sequence analyses.
Suggesting to add data of microscopic parasite detection because authors described parasite detection in lines 174 and 175.
Table 1
Suggesting to change to “18 (5.81%)”.
Discussion
Many short sentences were listed mechanically. I suggest to make several paragraphs.
Comments on the Quality of English Language
Needs reversion for introduction and discussion.
Author Response
Thank you very much for taking the time to review this manuscript. Please find the detailed responses below and the corresponding revisions/corrections highlighted/in track changes in the re-submitted files.
Please see pdf.

Reviewer 2 Report
Comments and Suggestions for Authors
This manuscript presents valuable and interesting data regarding the DNA prevalence of B. caballi in horses in Romania. The authors have made considerable efforts to present their research and support their findings. However, extensive editing is required before the manuscript can be considered for publication.
As a general comment, the authors should carefully review and edit their manuscript. Major improvements in English language usage are necessary. Below are specific comments for suggested changes, although this is not an exhaustive list of revisions needed throughout the manuscript.
Line 12: Use “detect B. caballi DNA” instead of “identify B. caballi,” as the authors did not perform microscopy to identify the parasite within the erythrocytes.
Line 16: Is “endemicity” for B. caballi in the regions studied concluded from the approximately 6% DNA prevalence found in this study? Moreover, is the parasite “endemic”, or are the areas “endemic”? In any case, the term “enzootic” is the most appropriate, and the authors should explain how this is concluded from their results.
Line 21: Keywords should typically be listed in alphabetical order.
Line 40-42: What are the “digenetic” methods? The authors should clarify when immunological methods like ELISA and IFAT are useful (e.g., which stage of infection) and explain why molecular methods are preferred in certain cases, as well as microscopy.
Lines 46-47: Is a PCR-positive result always conclusive about a recent or active infection? This sentence is not clear.
Line 49 and elsewhere throughout the manuscript: When introducing a short name (e.g., RAP-1) for the first time, it should be written in full with the abbreviation in parentheses. The authors should correct this.
Lines 51-54: Rewrite for clarity.
Lines 66-72: Please define that in parentheses you mention the number of samples collected from each county because this is not clear. Maybe something like “n=…”
Line 77 and elsewhere throughout the manuscript: The authors should provide a full description of the company e.g., (Sigma-Aldrich, St Luis, MO, USA)
Line 81-82: “DNA levels vary depending on the quality and number of cells in the sample, so for whole blood, optimal levels are between 8 and 50 ng/μL of blood [16]” This sentence is not needed here.
Line 88: delete “respectively”
Lines 92-94: The primers’ and probe’s sequences can be added to the previous paragraph when mentioning them or in a table.
Line 95: Please clarify the relevance of mentioning AgPath-ID™ One-Step RT‒PCR Reagents here, as it seems unrelated to the study.
Lines 108-109: Please correct this sentence. Positive and negative samples had a Ct value ≤ 40?? Also, explain how you defined this threshold.
Line 111: “five” instead of “5”.
Lines 111-120: Why did the authors choose to sequence such a small region of 95 bp? Are these sequences submitted to GenBank? What are the accession numbers?
Lines 129-132: his paragraph should be rewritten for clarity. Additionally, the variables “sex” and “age” were not compared with molecular results. The authors should state that they statistically analyzed differences in PCR positivity among age and sex groups.
Line135: “B.caballi DNA”
Lines 140-152: This section is overly verbose and does not critically present the results. It repeats information available in Table 1.
Table 1: “Equine species” instead of “type of equine” and “sex” instead of “gender”
Why did the authors use “*” for all the footnotes? They can simply write e.g., N: number of positive samples etc. Additionally, I do not understand where the p-values and the number of localities with positive samples are in the Table and why animal species (horse), age and sex are mentioned in the legend of the Table as “risk factors”.
Line 157: From which countries are these accession numbers?
Discussion section: This section is disorganized and lacks focus. I recommend that the authors reorganize it, emphasize relevant points pertaining to their study, avoid generalizations, and correct grammatical issues. Some sentences or paragraphs (e.g., lines 231-235) appear irrelevant.
The images provided lack meaningful context. The agarose gel photo should include information about the ladder, positive and negative controls, and tested samples. For the amplification curves, clarify what the reader is meant to interpret, as the real-time PCR was still running when the photo was taken. The threshold and Ct values for the gene under investigation and the internal control, the positive and negative controls ect are not visible.
Comments on the Quality of English LanguageImprovements in English language usage are necessary.
Author Response

(The authors gave the same response as above.)

Reviewer 3 Report
Comments and Suggestions for Authors
Please see the attached file.

Please see the attached file.
Author Response

(The authors gave the same response as above.)

Reviewer 4 Report
Comments and Suggestions for Authors
The manuscript presents an important epidemiological study regarding the infection with B. caballi in horses, in a region where such data indeed are lacking. Overall the manuscript is well written, but in my opinion it needs some corrections, as mentioned bellow:
Line 17: any symptoms resembling to clinical babesiosis
Line 33-35: the phrase needs to be revised
Line 55: Babesia caballi, not abbreviation of the genus, because is the beginning of the phrase
Line 71: the number of horses from Satu Mare (4) and Bistrița Năsăud (2) is too small to evaluate the prevalence of infection from these regions. In my opinion better to remove these horses from the study.
Line 143-144: the phrase contains repetitive information from the previous phrase, please change with “The youngest positive horse was 20 months, and the oldest was 277 months old.”
Line 147: the horses coming from the county of Gorj had the highest infection
Line 148: remove 14/91 between brackets, it was mentioned in the previous phrase
Please review the details from table 1, I don’t see the p values there. Also change the the prevalence value to 5.81 if this was mentioned before, not 5.80
Line 155 – number of localities with positive samples??? This is info not provided in the table
Line 176 – the susceptibility of horses to B. caballi infection is clear, this is not a result of this study, please remove “the susceptibility of horses to this parasite”
Line 185: add here your reference 28, a study from France where the authors detected a prevalence of 6.3% using real time PCR from horses blood
Line 187: reference 23 doesn’t provide any info about the prevalence of B. caballi in Hungary
Line 191-192: check the reference, the discussion is different than the reference provided
Line 200: cut the reference 28, it is not about the context, the study was made in France.
Line 213-214: please check, there are some errors
Line 221-226: please review the discussions between the mentioned lines. Why the authors mention a higher prevalence in Romania for B. caballi infection? The present study provide 5.80% prevalence for B. caballi (by PCR) and reference 47 provide 12.84 % for Theileria (by ELISA).
Line 244-248: better to split the content in 2 phrases. Remove “clinical” from “clinical veterinarians” or use “veterinary practitioners”, also better to cut “similar” from line 246
Comments on the Quality of English LanguageEnglish must be improved, some phrases are difficult to understand.
Author Response

(The authors gave the same response as above.)

Round 2
Reviewer 1 Report
Comments and Suggestions for Authors
I still have following concerns.
What is simple summary? What is a difference between simple summary and abstract?
Select better one.
“Introduction” is still segmented. Again, I suggest to make two or three paragraphs such as general information of equine babesiosis including epidemiology of world-wide and diagnostic methods, and an aim of present study.
“Discussion” is too long and still segmented. I suggest to make short with several paragraphs by focusing important findings.
Comments on the Quality of English LanguageSummarize introduction and discussion in several paragraphs.
Author Response
Dear Reviewer, thank you very much for taking the time to review this manuscript. We appreciate your attention to our work and your feedback for improving the manuscript.
Please find the detailed responses in the manuscript and pdf.

Reviewer 3 Report
Comments and Suggestions for Authors
Please see the attachment

The manuscript requires an extensive language review and it should be done before the submission. As a reviewer, I have to assess the text presented, and the promise "The revised manuscript will be sent for editing" is not enough.
Author Response

(The authors gave the same response as above.)

Round 3
Reviewer 1 Report
Comments and Suggestions for Authors
I have no further suggestions.
Author Response
Dear Reviewer,
I am most grateful to you for taking the time to review this manuscript and for providing me with your final recommendation.
Reviewer 3 Report
Comments and Suggestions for Authors
Dear authors,
as the editor/journal policy didn't provide you enough time to proceed with major revisions, I have to leave the final decision to the editor.
I understand from your report, that "you do exactly the same as the others" (relying on qPCR with no further validation or characterization), which I consider insufficient for piroplasmids, especially in times of growing evidence that relying on exclusively 18S marker leads to underestimation of species variability.
Good luck with your future research.
Comments on the Quality of English Language
OK
Author Response
Dear reviewer,
Thank you once again for taking the time to review this manuscript and we would like to inform you that we have done our best to take your suggestions into account to improve the manuscript and make it publishable. With the hope that you will agree to its publication, we send you our best regards.